# In Vitro Toxicity Evaluation of Carrageenan on Cells and Tissues of the Oral Cavity

**DOI:** 10.3390/md20080502

**Published:** 2022-08-03

**Authors:** Babatunde Y. Alli, Akshaya Upadhyay, Yuli Zhang, Belinda Nicolau, Simon D. Tran

**Affiliations:** Faculty of Dental Medicine and Oral Health Sciences, McGill University, 2001 McGill College Avenue, Montreal, QC H3A 1G1, Canada; akshaya.upadhyay@mail.mcgill.ca (A.U.); yuli.zhang@mail.mcgill.ca (Y.Z.); belinda.nicolau@mcgill.ca (B.N.); simon.tran@mcgill.ca (S.D.T.)

**Keywords:** carrageenan, human papillomavirus, oral mucosa, dental materials, mouthwash

## Abstract

Carrageenan is a highly potent anti-human papillomavirus (HPV) agent with the potential for formulation as a mouthwash against oral HPV infection. However, its toxic effect on tissues of the oral cavity is currently unknown. This study aims to evaluate the safety of carrageenan on human cells and tissues of the oral cavity. Human salivary gland cells and reconstructed human oral epithelium (RHOE) were used for this in vitro study. The cells were subjected to 0.005–100 µg/mL of carrageenan for 4, 12, and 24 h in quadruplicate. RHOE were exposed to 100 µg/mL of carrageenan for 24 h in triplicate and stained with hematoxylin/eosin for histological analyses. All experiments had saline and 1% sodium dodecyl sulphate (SDS) as negative and positive controls, respectively. Carrageenan tissue toxicity was evaluated using a 3-(4,5-dimethylthiazol-2-yl)-2,5-diphenyltetrazolium bromide (MTT) assay to quantify cell viability. Tissue toxicity was further evaluated histologically by an oral pathologist to assess morphological changes. Our data showed that carrageenan did not significantly decrease cell and tissue viability when compared to the positive control. The histological evaluation of the RHOE also showed no loss of viability of the carrageenan-treated sample compared to untreated tissue. In contrast, 1% SDS-treated RHOE showed extensive tissue destruction. Our experiments suggest that carrageenan is safe for use in the oral cavity.

## 1. Introduction

Human papillomaviruses (HPV) are one of the most common sexually transmitted infections in humans [1,2]. HPV causes a myriad of anogenital cancers, including anal and cervical cancer, and an increasingly larger share of oropharyngeal cancers (OPCs) in North America [3,4]. In decades past, tobacco smoking was the primary aetiological agent in OPCs. However, there has been a fundamental aetiological role reversal in recent years in North America in favour of HPV [4]. With an increase in HPV-positive OPCs (HPV-OPC) by as much as 225% since the 1980s [5,6], these cancers have silently reached epidemic proportions in North America, surpassing the annual incidence and mortality of cervical cancers—the most well-known HPV-related cancer—in North America [1,7,8].

Carrageenan is a marine product that is extracted from red seaweed [9]. It is a strong antiviral agent with demonstrated activity against HPV. One major mechanism by which HPV infects the cell is through its initial interaction with the cell surface glycosaminoglycans, such as heparan sulphate [10,11,12]. This observation is the basis for the demonstrated HPV inhibitory activities of sulphated polysaccharides, such as heparin, cellulose sulphate, and dextran sulphate [12,13,14]. Essentially, these sulphated polysaccharide compounds block HPV infection by mimicking cell surface heparan sulphate to compete for the HPV binding sites for cellular attachment. In addition to this mechanism, they also exert a post-attachment effect by continually blocking the infectivity of the cell-bound viruses [9]. Carrageenan is a class of sulphated polysaccharides with a remarkable structural similarity to these aforementioned sulphated polysaccharides (Figure 1), conferring the same anti-HPV activities. In fact, carrageenan is thousands-fold more potent than heparin in inhibiting HPV infection in vitro [9].

Based on these, the anti-HPV activity of carrageenan has been extensively examined in the literature and has shown great promise as a microbicide—mainly in sexual lubricant form—against anogenital HPV infection for the prevention of cervical cancers (reviewed in [15]). Conspicuously absent, however, is the evaluation of carrageenan against oral HPV infection to alleviate the increasing burden of HPV-OPC. Testing carrageen, for example, as a mouthwash formulation, against oral HPV infections in a clinical trial is a promising yet unexplored research endeavour [15].

Notwithstanding the potential efficacy of carrageenan for the prevention and/or treatment of oral HPV, there exists longstanding safety concerns about its use in humans [16,17,18,19]. Indeed, a recent clinical trial, the LIMIT-HPV Study, testing a carrageenan-based sexual lubricant against anal HPV infections among men who have sex with men was stopped early due to adverse events and a lack of efficacy [20]. Although a preliminary analysis of another trial, the CATCH Study, by the same research group found that the lubricant reduces the risk of vaginal HPV infection in women, there was a higher rate of adverse effects in the carrageenan arm [21]. Therefore, for ethical and safety reasons, it is essential to evaluate the toxicity of carrageenan in the oral cavity before embarking on a clinical trial for oral HPV prevention. In this study, we conduct a series of in vitro experiments to test the effect of relevant concentrations of carrageenan on cells and tissues of the human oral cavity.

## 2. Results

Across the eight tested concentrations, there was no statistically significant difference in human normal salivary gland-SV40 transformed-acinar cell (NS-SV-AC) viability when exposed to either carrageenan or saline control at 4, 12, and 24 h, except for 1 µg/mL carrageenan at 24 h, showing lower viability than control (*p* = 0.029) (Figure 2). At 100 µg/mL, the highest tested concentration, the time to the toxicity of carrageenan was comparable to the negative saline controls over 24 h (Figure 3).

Figure 4 shows that at 24 h, the viability of RHOE tissues exposed to carrageenan was comparable to saline control and statistically significantly higher than the positive control (global *p*-value = 0.0025). The pairwise comparison shows no difference in viability between carrageenan and saline (*p* = 0.309), but a statistically significant difference in tissue viability between carrageenan and 1% SDS (*p* = 0.0074) and saline and 1% SDS (*p* = 0.0074). This was confirmed via histological evaluation (Figure 5). The histological sections of the carrageenan-treated tissue and the untreated control (Figure 5; top and middle, respectively) show non-keratinized stratified squamous epithelium of varying thickness with no abnormal nuclear features, no obvious tissue damage, and a comparable morphology. In contrast, the section of tissue treated with 1% SDS had extensive tissue destruction with the basal layer detached from the polycarbonate filter, and only thin fragmented strands of an otherwise unrecognizable tissue can be appreciated (Figure 5, bottom).

## 3. Discussion

In this study, we evaluated the toxicity of carrageenan on the oral cavity in vitro using human salivary gland cells and reconstructed human oral mucosa tissue. Our experiments show that carrageenan is non-toxic to the oral cavity even at a concentration that is up to twenty-thousand-folds higher than its IC_50_ against HPV. Importantly, this in vitro assay uses the RHOE tissue model, which is standardized for the testing of toxicity of dental materials and has been shown to reliably predict the toxicity of oral care product formulations in humans [22]. This result adds to the evidence in favour of the safety of carrageenan [18,23]. 

Our result suggests that carrageenan is potentially safe on the cells and tissues of the oral cavity. Importantly, the LIMIT-HPV investigators postulated that the adverse effect of the tested lubricant is due to the hyperosmolarity of the gel and not specifically due to the activity of carrageenan [20]. This is based on previous sexual lubricant research that has shown that hyperosmolar lubricants induce rectal epithelial damage leading to noxious side effects and an increased risk of sexually transmitted infection [24]. Nevertheless, given the results of the LIMIT-HPV study and for safety reasons, the onus is on future studies to show that carrageenan evaluated alone is safe; hence, this study was conducted. Another salient point from these trials is that any potential side-effects of carrageenan and its severity might be organ/tissue specific. Although there were multiple study-related withdrawals in the carrageenan arm of the LIMIT-HPV (anus) [20], there were none in the CATCH study (vagina) [21]. This was attributed to the physiologic self-lubricating mechanisms of the vagina that is absent in the anus. Therefore, given that the oral cavity is bathed in saliva, and it is one of the most resilient parts of the human body to noxious external stimuli, it is reasonable to suggest that any side-effect of carrageenan will be minimal in human use in the oral cavity.

SDS (also known as Sodium Lauryl Sulphate) is a model irritant [25], and its use in this study as positive control is further informed by the fact that it is the most common foaming agent in toothpaste and mouthwashes where it is usually used at a concentration between 0.5% and 2% [26]. Despite this well-known toxicity of SDS and its clinically proven adverse effect in potentiating episodes of recurrent aphthous stomatitis [27] and burning mouth syndrome [28], SDS has not yet been displaced as the predominant detergent in personal oral care products. Our results show that carrageenan is much safer than the ubiquitous SDS on tissues of the oral cavity and suggests that the use of a carrageenan-based mouthwash will have little to no adverse effect in a clinical trial setting. 

There are two main independent objectives of toxicity testing [29]. One is to evaluate if a material will be safe under the condition of expected use, referred to as a safety evaluation [29]. Here, the exposure concentration is informed by the proposed use of the product. This is reflected in our tested range of carrageenan concentrations within which carrageenan is still a highly potent anti-HPV agent [9], but above which it exhibits rheological properties that make it unfavourable for a mouthwash formulation. The other objective of toxicity testing is to determine the safe upper limit of exposure to the test material, referred to as a hazard evaluation [29]. This is beyond the scope of our study, and our results should be interpreted with that in mind. 

In conclusion, carrageenan is a strong anti-HPV material that has shown promise in microbicidal sexual lubricant formulations against genital HPV infection. Formulating it as a mouthwash for the prevention and/or clearance of an oral HPV infection is a promising research proposition to reduce the burden of HPV-OPC. Our study shows that carrageenan is potentially safe for use for this purpose in the oral cavity, especially when compared to SDS. However, despite this promising safety in vitro, we still recommend that future clinical trials testing a carrageenan-based agent in the oral cavity proceed cautiously by first piloting their protocols to obtain preliminary safety data in humans. 

## 4. Materials and Methods

### 4.1. Preparation of the Test Agent

Iota-carrageenan, the most potent of the three main forms of carrageenan [9], was obtained from Sarda Biopolymers, Mumbai, India (Batch number: SBPL-030/21-22). The powder was then prepared into an aqueous 0.05% stock solution (0.5 mg/mL) with normal saline as the vehicle. The solution was then serially diluted to make the following test concentrations (in µg/mL): 0.005, 0.01, 0.1, 1, 5, 25, 50, and 100. The minimal end of this concentration series (0.005 µg/mL) was selected based on the 50% inhibitory concentration (IC_50_) of carrageenan against HPV16, as reported in Buck et al. [9]. At this concentration, carrageenan is thousands-fold more potent as an inhibitor of HPV16 than heparin, which is the model papillomavirus inhibitor in the laboratory. The selected maximum concentration of 100 µg/mL represents a twenty-thousand-fold increase on the IC_50_, where any concentration within this range should provide a highly potent mouthwash application against HPV. 

### 4.2. Human Salivary Gland Cell Culture

A well-characterized human salivary cell gland acinar cell (NS-SV-AC) line was used. These cells were a gift from Prof. Masayuki Azuma, Tokushima University of Dentistry, Japan. The cells were plated in a 96-well plate and incubated with Dulbecco’s minimal essential medium supplemented with penicillin, streptomycin, and 10% fetal bovine serum at 5% CO_2_ and 37 ℃. The wells were then treated in quadruplicates with the test agent carrageenan with all the freshly prepared concentrations (0.005–100 µg/mL) with normal saline (also in quadruplicates) as negative controls. One quadruplicate of NS-SV-AC was treated with 1% sodium dodecyl sulphate (SDS) as a positive control. The treated NS-SV-AC cells were incubated for 4, 12, and 24 h before assessments for viability. We used the NS-SV-AC cells because they are epithelial cells adjacent to and similar in origin to the lining of the oral cavity, and our lab has extensive expertise in its cultivation and use.

### 4.3. Human Oral Epithelial Tissue Culture

Reconstructed human oral epithelial (RHOE) tissue obtained from Episkin Laboratories, Lyon, France, was used for tissue toxicity testing. The RHOE tissue model is a living, multilayered epithelial tissue devoid of stratum corneum that histologically resembles the human oral mucosa. They are produced on polycarbonate inserts in vitro from the transformed keratinocytes of the TR146 cell line [30,31]. The reconstructed tissue recapitulates most of the structural and functional features of the oral mucosa.

On arrival, the epithelium went through a 24 h maintenance period following the manufacturer’s instructions as follows: (i) the RHOE was removed from the shipping agarose with all excess agar removed; and (ii) individual tissues were then placed individually in a 6-well plate pre-filled with a maintenance medium and incubated at 37 °C, 5% CO_2_, and saturated humidity. After 24 h, the maintenance medium was changed, and the tissues were topically exposed to 150 µL of 100 µg/mL concentration of carrageenan in triplicate. Normal saline and 1% SDS were used as negative and positive controls, respectively. All treated tissues were incubated at 37 °C, 5% CO_2_, and saturated humidity for 24 h. 

### 4.4. Cell and Tissue Viability 

Following the treatment periods, tissue viability was evaluated by quantitating cellular mitochondrial dehydrogenase activities using the 3-(4,5-dimethylthiazol-2-yl)-2,5-diphenyltetrazolium bromide (MTT) assay [32]. The MTT assay is a colourimetric technique used to measure cellular metabolic activities as an indicator of cell viability [32]. This is based on the observation that metabolically active cells reduce yellow tetrazolium salts in solutions to form purple formazan crystals [32,33]. The crystals are then dissolved in a solubilization solution, and the ensuing coloured solution can be quantified by a spectrophotometer: the darker the solution, the greater the number of viable cells. 

At 4, 12, and 24 h, the test agents were removed from the NS-SV-AC cell cultures and replaced with 10 µL MTT solution and incubated for 3 h. Afterward, the MTT solution was removed and replaced with dimethyl sulfoxide (DMSO) to dissolve the crystals at room temperature for 5 min. Then, the 96-well plate was transferred to the Microplate Reader, and the optical density (OD) was read at 570 nm.

Further, following the 24 h test period, the RHOE tissue cultures were washed with PBS, then placed in a 300 µL of 0.5 mg/mL MTT solution and incubated at 37 ℃ for 3 h. After 3 h of incubation, MTT crystals were extracted by transferring the culture into 1.5 mL isopropanol for 1.5 h at room temperature. Finally, 200 µL of the extracted solution from each well was transferred to a 96-well plate, and the OD was measured at 570 nm using an EL800 Universal Microplate Reader. 

### 4.5. Histological Examination

Following the manufacturer’s instructions, the RHOE tissue was cut off the plastic insert with the polycarbonate filter intact. The tissue was fixed in 10% formalin and underwent standard histopathological processing for hematoxylin and eosin staining. The processed slides were assessed for toxicity by evaluating cellular and tissue morphological changes. 

### 4.6. Statistical Analysis

Statistical analysis was conducted using R software [34]. The mean and standard deviation of the OD was taken from duplicate to quadruplicate experiments. Tissue viability was expressed as a percent of the negative culture as OD test/OD negative control ×100. We used the Wilcoxon test to make a two-way comparison between carrageenan and saline control for the NS-SV-AC assays and the Kruskal–Wallis test for the three-way comparison between carrageenan, negative control, and positive control for the RHOE tissue cultures with pairwise Wilcoxon test with Bonferroni correction for individual comparisons. The confidence level was set at 95%, and the statistical significance level was set at 0.05

## Figures and Tables

**Figure 1 marinedrugs-20-00502-f001:**
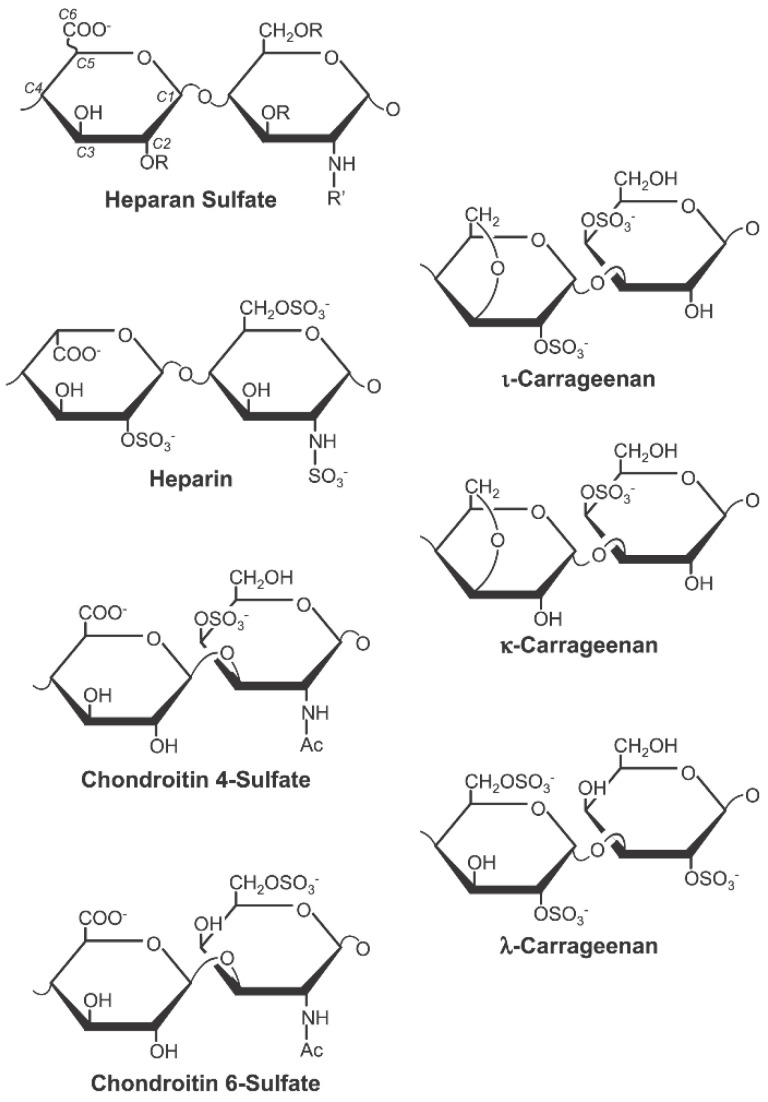
The three main forms of carrageenan (iota, kappa, and lambda) show structural similarities with the major sulphated polysaccharide types (Figure reused from Buck et al. citation [9] under Creative Commons Attribution Licence).

**Figure 2 marinedrugs-20-00502-f002:**
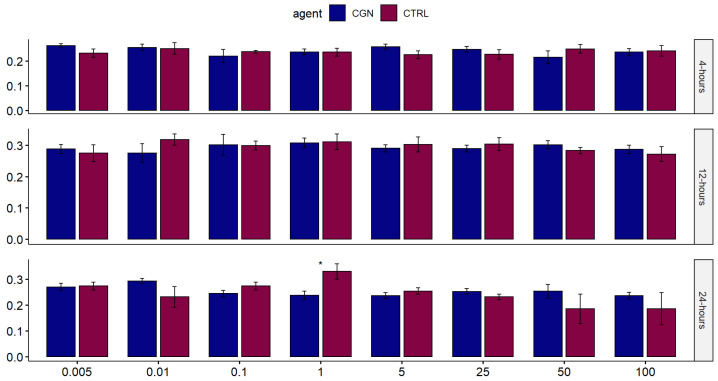
Comparison of the effect of different concentrations of carrageenan on salivary gland cells against saline control over 24 h. Each bar represents the mean value from quadruplicate samples (* signifies statistical significance).

**Figure 3 marinedrugs-20-00502-f003:**
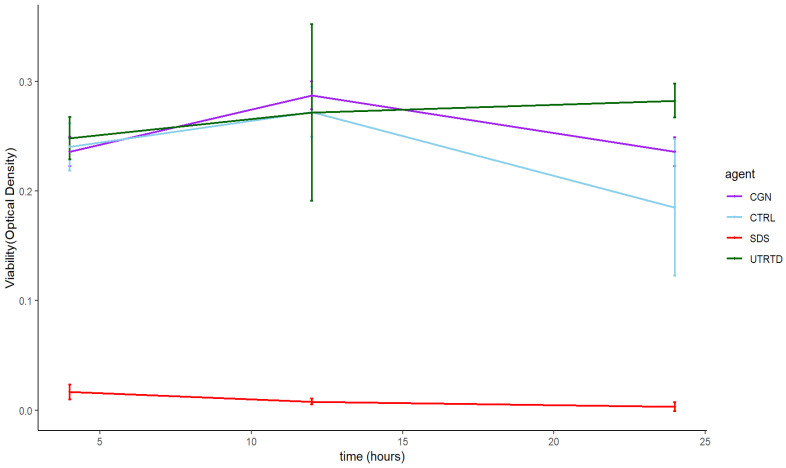
Time to the toxicity of 100µg/mL carrageenan (CGN) on salivary gland cells over 24 h compared to saline (CTRL), 1% SDS (SDS), and untreated cells (UNTRTD) controls.

**Figure 4 marinedrugs-20-00502-f004:**
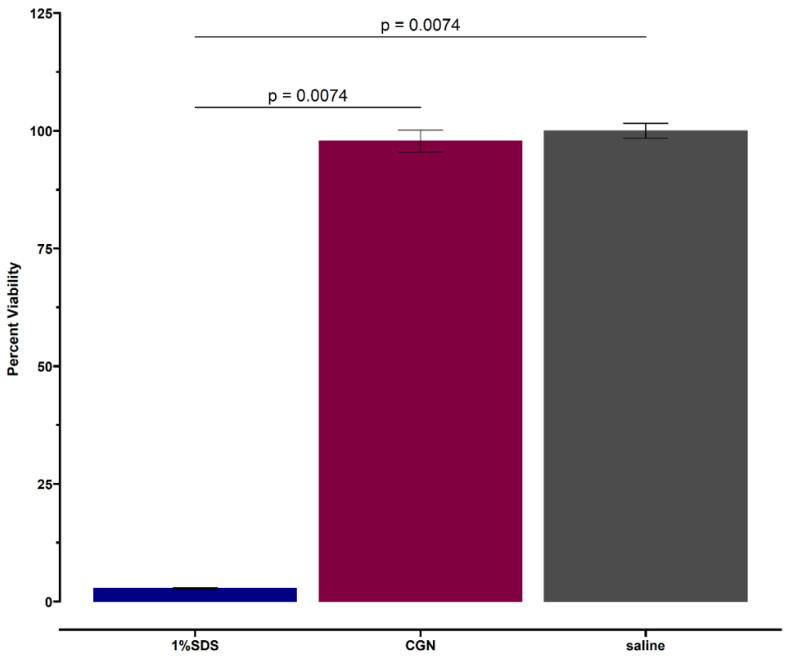
Twenty-four-hour viability of tissues exposed to carrageenan (CGN) with saline (negative) and 1% SDS (positive) controls. Each bar represents the mean value from duplicate samples.

**Figure 5 marinedrugs-20-00502-f005:**
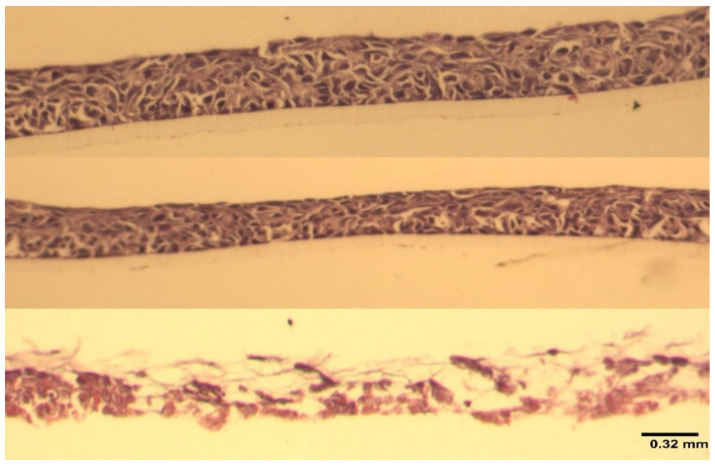
A composite figure of the corresponding histological sections from Figure 4. Top to bottom—carrageenan-treated (100 µg/mL), untreated control, positive control (1% SDS). 20× hematoxylin and eosin staining. Image scale: 0.32 mm.

## Data Availability

Data available on request to the corresponding author.

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
