# Peer review of "In Vitro Toxicity Evaluation of Carrageenan on Cells and Tissues of the Oral Cavity"

_marinedrugs, 2022, doi:10.3390/md20080502_

Round 1

Reviewer 1 Report

The work by Alli et al., evaluates the toxicity of carrageenan on cells and tissues of the oral cavity, using the MTT assay on the Human salivary gland cells and histological analysis of reconstructed human oral 11 epithelium (RHOE). Although the subject of the investigation is relevant, I found that the work could be substantially improved. The points that in my opinion need to be addressed are: 

1- The study lacks functional inhibition experiments supporting that carrageenan (at the concentration used in the toxicity experiments) inhibits the binding of the HPV virus to oral cavity cells and the subsequence infection;

2- The use of MTT- Although the MTT assay for cellular metabolic activity is almost ubiquitous in studies of cell toxicity, it is commonly applied and interpreted erroneously. Considerations in performing MTT assay:

- Does the MTT assay represent what you aim to measure? 

- Have the following parameters been considered in the study? 

1- Cell seeding number and density,

2- MTT concentration,

3- MTT incubation time,

3- Culture conditions (such as culture media type, presence of serum, and phenol red in the media, etc.)

4- Assay measurement by either optical or chemical interference? Are any effects of tested treatments considered that could affect the final OD measurements in a direct or indirect way? 

- MTT uptake and/or extrusion (e.g., cell membrane permeability/integrity) 

- Cell number (e.g., proliferation) 

- Cell metabolism (e.g., chemo/radio-induced senescence-like phenotype) 

- Cell secretome (e.g., chemo/radio-induced senescence-associated secretory phenotype) 

- Background absorbance and scattering 

- Abiotic reduction of MTT.

The quality of the figures needs to be improved, especially the histology analysis.

Has the pH of the solution been considered, since it can have effects on the carrageenan inhibitory effect?

I believe the work could be significantly improved if these points were discussed in the manuscript.  

Author Response

The study lacks functional inhibition experiments supporting that carrageenan (at the concentration used in the toxicity experiments) inhibits the binding of the HPV virus to oral cavity cells and the subsequence infection

We thank the reviewer for this comment. However, given the existing extensive literature on the demonstrated effects of Carrageenan on HPV (reviewed extensively in Laurie et al doi: 10.1097/OLQ.0000000000001363), we did not consider this germane to achieving our primary aim. Yes, we had to make a decision on the concentrations to be tested with the primary concern that these concentrations have to be relevant to HPV and be highly efficacious against the virus. Therefore, as we described in the manuscript, we based this concentration on the landmark work by Buck et al where they showed that even as small a concentration as 0.005 µg/ml Carrageenan was highly active against HPV and this was used as our minimum concentration while our maximum concentration was an arbitrary concentration that is twenty-thousand-fold higher than this (100 µg/ml). This maximum concentration is higher or the same as many of the highly active experiments reviewed in Laurie et al. In summary, in the absence of evidence from the literature, we would have been compelled to run these inhibition assays to arrive at our tested concentrations, however, there is an abundance of literature on the efficacy of carrageenan at our tested concentrations that renders such inhibition assays unnecessary.

2- The use of MTT- Although the MTT assay for cellular metabolic activity is almost ubiquitous in studies of cell toxicity, it is commonly applied and interpreted erroneously. Considerations in performing MTT assay:

- Does the MTT assay represent what you aim to measure? 

- Have the following parameters been considered in the study? 

1- Cell seeding number and density,

2- MTT concentration,

3- MTT incubation time,

3- Culture conditions (such as culture media type, presence of serum, and phenol red in the media, etc.)

4-Assay measurement by either optical or chemical interference? Are any effects of tested treatments considered that could affect the final OD measurements in a direct or indirect way? 

- MTT uptake and/or extrusion (e.g., cell membrane permeability/integrity) 

- Cell number (e.g., proliferation) 

- Cell metabolism (e.g., chemo/radio-induced senescence-like phenotype) 

- Cell secretome (e.g., chemo/radio-induced senescence-associated secretory phenotype) 

- Background absorbance and scattering 

  • Abiotic reduction of MTT.

We thank the reviewer and appreciate the reviewer's concern about the MTT assay. However, all of the issues raised here were addressed in our MTT experiments with many items stated in our extensive method sections on the MTT. Also, the MTT assay for the tissue was performed following the tissue manufacturer's recommendation and description. Furthermore, our lab has extensive experience with MTT assays on the NSSVAC cells which we conduct routinely with a standardized number of cells seeded in each well-plate based on a standard curve.  All background absorbance is accounted for during the reading of the well plates and our experiments are conducted in triplicates to quadruplicates with negative and positive controls in paired experiments or higher. 

The quality of the figures needs to be improved, especially the histology analysis

We have now improved the quality of many of the figures based on this comment and other reviewers' comments. 

Reviewer 2 Report

My main issue with this manuscript is how certain figures are displayed.  Figure 4A seems to be the best way to display the cell viability data, with cell viability displayed as % viability on the y-axis, while the concentrations of carrageenan (or control) are given on the x-axis.  For Figure 2, I would advise for switching the axes and the style of graph (I would recommend using a bar graph); I would further advise differentiating the different treatment times with different bar styles and differentiating between control-treated and carrageenan-treated data with contrasting colors.   For Figure 3, I would also advise similar changes as with Figure 2, and I would also advise using easily-discernable colors for the treatments - it was hard for me to discern between control and carrageenan-treatment groups.  

Another issue I have is the the dearth of descriptors with the figure legends and in the results section.  For example, while the number of experiments is listed in the text, it would help the reader to quickly scan the legend and see how many replicates (technical and actual, with actual experiments being listed as "N=") the figure data represents.  Another example is when cell lines such as NS-SV-AC are first introduced into the text, it would help the reader to first identify them as a human salivary gland cell line and then keep the further elaboration of the cell line in the Materials and Methods section.

Last major thought - has there been any consideration of seeing what the cell viability of your carrageenan treatment scheme would look like in a + or - HPV cell line, such as head and neck squamous cell carcinoma (HNSCC)? 

Minor corrections:

I may be mistaken on this, but I don't think carrageenan is a brand or proper name, so it shouldn't be constantly capitalized throughout the text when it is not the beginning word of a sentence. 

Lines 13-14: "quadruplicates" should be "quadruplicate".  Also, reword sentence: "Triplicate RHOE were exposed to 100 µg/ml of Carrageenan for 24 hours and stained with haematoxylin/eosin for histological analyses" to "RHOE were exposed to 100ug/ml of carrageenan for 24 in triplicate...".

Line 15: "Sodium Dodecyl Sulfate" should be lower case.

Lines 16-17: Reword sentence: "The toxicity of cells and tissues was evaluated using a 3-(4,5-dimethylthiazol-2yl)-2,5-diphenyltetrazolium bromide (MTT) assay to quantify cell viability." to "Carrageenan tissue toxicity was evaluated...".

Lines 18-20: Reword sentence: "The MTT assay showed no loss of cell and tissue viability for Carrageenan when compared to the negative (vehicle) control and a statistically significant better tissue viability when compared to the positive control" to "Our data showed that carrageenan did not significantly decrease cell and tissue viability...".

Lines 22-23: Can omit "In contrast, 1% SDS-treated RHOE showed extensive tissue destruction." sentence from the abstract.

Line 29: Change "myriad form" to "myriad of cancers including...".

Line 49: Change "in-vitro" to "in vitro". 

Lines 49-50: Figure 1 looks compressed, is there a way to get a higher-res image from the original source?

Line 52: Change "et al." to "et al".

Line 71: Change "conduct" to "conducted".

Lines 108, 145, and 148: Change "IC50" to "IC50".

Line 145: Change "et al" to "et al".

Author Response

My main issue with this manuscript is how certain figures are displayed.  Figure 4A seems to be the best way to display the cell viability data, with cell viability displayed as % viability on the y-axis, while the concentrations of carrageenan (or control) are given on the x-axis.  For Figure 2, I would advise for switching the axes and the style of graph (I would recommend using a bar graph); I would further advise differentiating the different treatment times with different bar styles and differentiating between control-treated and carrageenan-treated data with contrasting colors.   For Figure 3, I would also advise similar changes as with Figure 2, and I would also advise using easily-discernable colors for the treatments - it was hard for me to discern between control and carrageenan-treatment groups.  

We thank the reviewer for the thoughtful suggestions for the figure presentations. We have now implemented many of these suggestions for the new Figure 2 and Figure 3.

Another issue I have is the the dearth of descriptors with the figure legends and in the results section.  For example, while the number of experiments is listed in the text, it would help the reader to quickly scan the legend and see how many replicates (technical and actual, with actual experiments being listed as "N=") the figure data represents.  Another example is when cell lines such as NS-SV-AC are first introduced into the text, it would help the reader to first identify them as a human salivary gland cell line and then keep the further elaboration of the cell line in the Materials and Methods section.

We have now fully defined the NSSVAC cells at the first mention, thereby identifying them as human salivary gland cells. We also added the replicate information to the figures.

Last major thought - has there been any consideration of seeing what the cell viability of your carrageenan treatment scheme would look like in a + or - HPV cell line, such as head and neck squamous cell carcinoma (HNSCC)?

We didn't consider testing the HNSCC cell line since our primary idea was to evaluate the viability of "normal" tissues; i.e., as similar as possible to a healthy human oral cavity that might be encountered in a clinical trial scenario albeit in vitro

I may be mistaken on this, but I don't think carrageenan is a brand or proper name, so it shouldn't be constantly capitalized throughout the text when it is not the beginning word of a sentence. 

Lines 13-14: "quadruplicates" should be "quadruplicate".  Also, reword sentence: "Triplicate RHOE were exposed to 100 µg/ml of Carrageenan for 24 hours and stained with haematoxylin/eosin for histological analyses" to "RHOE were exposed to 100ug/ml of carrageenan for 24 in triplicate...".

Line 15: "Sodium Dodecyl Sulfate" should be lower case.

Lines 16-17: Reword sentence: "The toxicity of cells and tissues was evaluated using a 3-(4,5-dimethylthiazol-2yl)-2,5-diphenyltetrazolium bromide (MTT) assay to quantify cell viability." to "Carrageenan tissue toxicity was evaluated...".

Lines 18-20: Reword sentence: "The MTT assay showed no loss of cell and tissue viability for Carrageenan when compared to the negative (vehicle) control and a statistically significant better tissue viability when compared to the positive control" to "Our data showed that carrageenan did not significantly decrease cell and tissue viability...".

All done.

Lines 22-23: Can omit "In contrast, 1% SDS-treated RHOE showed extensive tissue destruction." sentence from the abstract.

We thank the reviewer for the suggestions. However, we are of the opinion that this contrast information is important in the abstract so we are leaving this sentence in place.

Line 29: Change "myriad form" to "myriad of cancers including...".

Line 49: Change "in-vitro" to "in vitro". 

All done.

Lines 49-50: Figure 1 looks compressed, is there a way to get a higher-res image from the original source?

We have downloaded the best resolution available from the journal website and included it in the new revised manuscript. However, the Word document could still make it look stretched.

Line 52: Change "et al." to "et al".

Line 71: Change "conduct" to "conducted".

Lines 108, 145, and 148: Change "IC50" to "IC50".

Line 145: Change "et al" to "et al".

All done

Reviewer 3 Report

The goal of this work was to assess the safety and efficacy of carrageenan as an additive to a mouthwash to combat human papillomavirus (HPV) infection. The authors employed in vitro models of human salivary gland cells and reconstructed human oral epithelium (RHOE), exposing them to 100ug/ml of carrageenan for 4, 12 and 24 hours and assessing damage by H&E staining followed by pathological scoring and by MTT assay. Overall, they found that carrageenan did not cause cell or tissue damage and therefore they conclude that carrageenan is safe for use in the oral cavity.

Overall, the manuscript is well written and clearly presented. However, I do not believe that the amount of work warrants publication as a full article. I believe it would be better suited to a short communication format.

My main thought was regarding previous studies of carrageenan as an ingredient of lubricants. The authors quote two trials – LIMIT-HPV and CATCH. In both studies the use of carrageenan was associated with adverse effects. This needs to be discussed more. I believe that more justification is needed as to why are the authors now studying carrageenan despite two clinical trials showing adverse effects. What justification do they have that use in the oral cavity will not present the same issues? Have previous studies investigated carrageenan in in vitro models of these tissues/cells? Have they also revealed promising results? If so, why do the authors believe that carrageenan will be suitable for oral use? 

This study would also be strengthened by the use of an HPV infection model to determine the effects of carrageenan in HPV infection in these cells. 

Minor comments;

Figure 1 is stretched and should be fixed.

Figure 2 – suggest moving p values so they do not overlap the bar graph lines

Figure 4A it is difficult to read the text

Line 69 – suggest using another term - research waste is not very specific.

Line 75 - NS-SV-AC is not introduced previously. It should be explained here.

Author Response

Overall, the manuscript is well written and clearly presented. However, I do not believe that the amount of work warrants publication as a full article. I believe it would be better suited to a short communication format.

We thank the reviewer for this comment. Indeed, we initially wrote the manuscript as a short communication (and the current version could probably still fit into that format due to its relatively low word count). However, the journal does not have any specific formatting for short communications which is why it is written in the present format.

My main thought was regarding previous studies of carrageenan as an ingredient of lubricants. The authors quote two trials – LIMIT-HPV and CATCH. In both studies the use of carrageenan was associated with adverse effects. This needs to be discussed more. I believe that more justification is needed as to why are the authors now studying carrageenan despite two clinical trials showing adverse effects. What justification do they have that use in the oral cavity will not present the same issues? Have previous studies investigated carrageenan in in vitro models of these tissues/cells? Have they also revealed promising results? If so, why do the authors believe that carrageenan will be suitable for oral use? 

We thank the reviewer for these thoughtful questions. To largely respond to this, we have added the following paragraph to the discussion section. 

"Our result suggests that carrageenan is potentially safe on the cells and tissues of the oral cavity. Importantly, the LIMIT-HPV investigators postulated that the adverse effect of the tested lubricant is due to the hyperosmolarity of the gel and not specifically due to the activity of carrageenan. This is based on previous sexual lubricant research that has shown that hyperosmolar lubricants induce rectal epithelial damage leading to noxious side effects and an increased risk of sexually transmitted infection. Nevertheless, given the results of the LIMIT-HPV study and for safety reasons, the onus is on future studies to show that carrageenan evaluated alone is safe; hence, this study was conducted. Another salient point from these trials is that any potential side-effects of carrageenan and its severity might be organ/tissue specific. Although there were multiple study-related withdrawals in the carrageenan arm of the LIMIT-HPV (anus), there were none in the CATCH study (vagina). This was attributed to the physiologic self-lubricating mechanisms of the vagina which is absent in the anus. Therefore, given that the oral cavity is bathed in saliva, and it is one of the most resilient parts of the human body to noxious external stimuli, it is reasonable to suggest that any side-effect of carrageenan will be minimal in human use in the oral cavity."

This study would also be strengthened by the use of an HPV infection model to determine the effects of carrageenan in HPV infection in these cells.

We thank the reviewer for the suggestion and this might potentially represent a potential future research endeavour. However, our primary aim for this study was to simply evaluate the potential toxicity of carrageenan on normal cells and tissue in vitro.

Figure 1 is stretched and should be fixed

We have downloaded a higher resolution of the image from source and included it in the manuscript.

Figure 2 – suggest moving p values so they do not overlap the bar graph lines

We have now removed all of the stated values to add clarity to the graph (Since only one of the comparisons was significant and this is now indicated with an asterik)

Line 69 – suggest using another term - research waste is not very specific.

We have dropped this term. We came to the conclusion that research waste as we've used it is also an ethical issue (either on the funders' part or the participant's time wasted). Therefore, just stating "for ethical and safety reasons" alone is adequate.

Figure 4A it is difficult to read the text

This has been fixed, and the values are now legible on Figure A

Line 75 - NS-SV-AC is not introduced previously. It should be explained here.

We have now defined NS-SV-AC on first use.

Reviewer 4 Report

The authors present an original data on the effect of carrageenan on human salivary gland cells and reconstructed human oral epithelium to evaluate it's potential use as a mouthwash. I recommend the acceptance of the manuscript pending major revisions: 

1. The interpretation of the data obtained in the manuscript within the context of oral HPV infection depends on the genotype of the viral isolate used in the experiments which is not provided. Please specify the HPV genotype used and the origin of the viral species. Make sure that you comment on this issue in results and discussion.

Minor comments: 

. In section Introduction, lines 28: Considering the molecular diversity of human papillomaviruses, please correct the sentence and use plural "Human papillomaviruses (HPVs) are.....". 

2. Please cite the web site https://www.ncbi.nlm.nih.gov/pmc/articles/PMC6171710/ or the reference Doorslae et al as a reference of the International Society for the Taxonomy of Viruses after the first sentence. 

3. Please specify HPV genotypes relevant for oral medicine. 

4. Please provide the data on the manufacturer of microplate readers used in the study. 

Author Response

The interpretation of the data obtained in the manuscript within the context of oral HPV infection depends on the genotype of the viral isolate used in the experiments which is not provided. Please specify the HPV genotype used and the origin of the viral species. Make sure that you comment on this issue in results and discussion.

We did not test carrageenan against oral HPV in this manuscript. Our primary aim was to test carrageenan toxicity in health cells and tissues of the oral cavity in vitro. That said, carrageenan has been tested extensively against HPV in the literature and it has been found to be extremely potent against the virus.

. In section Introduction, lines 28: Considering the molecular diversity of human papillomaviruses, please correct the sentence and use plural "Human papillomaviruses (HPVs) are.....". 

Done

Please cite the web site https://www.ncbi.nlm.nih.gov/pmc/articles/PMC6171710/ or the reference Doorslae et al as a reference of the International Society for the Taxonomy of Viruses after the first sentence. 

Done

Please provide the data on the manufacturer of microplate readers used in the study. 

Done

Round 2

Reviewer 3 Report

Overall the manuscript is improved. Please note that figure 4A has some issues with the fonts and scaling that should be addressed.

Author Response

Thank you for the comment on Figure 4A.  We finally decided that for clarity, it was unnecessary trying to combine figures 4A and 4B into a single figure. We have now split both figures into Figure 4 and 5 respectively (Moreso, since the journal has no hard limits on figures). Figure 4 is now a lot clearer as a result.

Reviewer 4 Report

The authors responded to all questions from the reviewer and completed all requests. I recommend the study for publication in the present form. 

Author Response

We thank the reviewer for the comment.